# Countermovement Rebound Jump: A Comparison of Joint Work and Joint Contribution to the Countermovement and Drop Jump Tests

**Jiaqing Xu [1], Anthony Turner [1], Thomas M. Comyns [2], John R. Harry [3], Shyam Chavda [1] and Chris Bishop [1,\***

[1] London Sport Institute, Middlesex University, London NW4 4BT, UK; jx066@live.mdx.ac.uk (J.X.); a.n.turner@mdx.ac.uk (A.T.); s.chavda@mdx.ac.uk (S.C.)

[2] Department of Physical Education and Sport Sciences, University of Limerick, V94 T9PX Limerick, Ireland; tom.comyns@ul.ie

[3] Human Performance & Biomechanics Laboratory, Department of Kinesiology & Sport Management, Texas Tech University, Lubbock, TX 79409, USA; john.harry@ttu.edu

\* Correspondence: c.bishop@mdx.ac.uk

**Abstract:** The kinetic analysis of joint work and joint contribution provides practitioners with information regarding movement characteristics and strategies of any jump test that is undertaken. This study aimed to compare joint works and contributions, and performance metrics in the countermovement jump (CMJ), drop jump (DJ), and countermovement rebound jump CMRJ. Thirty-three participants completed 18 jumps across two testing sessions. Jump height and strategy-based metrics (time to take-off [TTTO], countermovement depth [CM depth], and ground contact time [GCT]) were measured. Two-way analysis of variance assessed systematic bias between jump types and test sessions ($\alpha = 0.05$). Reliability was evaluated via intraclass correlation coefficient [ICC] and coefficient of variation [CV]. Jump height and strategy-based metrics demonstrated good to excellent reliability (ICC = 0.82–0.98) with moderate CV ($\leq 8.64\%$). Kinetic variables exhibited moderate to excellent reliability (ICC = 0.64–0.93) with poor to moderate CV ($\leq 25.04\%$). Moreover, apart from TTTO ($p \leq 0.027$, effect size [ES] = 0.49–0.62) that revealed significant differences between jump types, CM depth ($p \leq 0.304$, ES = 0.27–0.32) and GCT ($p \leq 0.324$, ES = 0.24) revealed nonsignificant trivial to small differences between three jumps in both sessions. Finally, the negative and positive hip and knee works, and positive ankle contribution measured in the CMRJ showed significant differences from the CMJ and DJ ($p \leq 0.048$, $g \leq 0.71$), with no significant difference observed in other kinetic variables between the three jump actions ($p \geq 0.086$). Given the consistent joint works and joint contributions between jump types, the findings suggest that practitioners can utilize the CMRJ as a viable alternative to CMJ and DJ tests, and the CMRJ test offers valuable insights into movement characteristics and training suggestions.

**Keywords:** force platforms; kinetics; motion capture; reliability; jump strategy

## 1. Introduction

Jump actions appear frequently in many sports; thus, they represent a time-efficient means of assessing lower body explosive power capabilities [1–3]. Whilst a plethora of options are available, the countermovement jump (CMJ) and drop jump (DJ) represent useful and commonly applied methods of assessing slow and fast stretch-shortening cycle (SSC) mechanics, respectively [4,5]. However, testing large squads of athletes with both jumps naturally extends the duration of any given test session, which may not always be favorable when time is finite in elite sport settings [1]. In this instance, the countermovement rebound jump (CMRJ) that combines the characteristics of both the CMJ and DJ might be a viable alternative and allow practitioners access to comparable information [6]. The

CMRJ requires athletes to perform an initial maximal effort CMJ (i.e., CMRJ1), immediately followed by a second, reactive vertical drop jump (i.e., CMRJ2) which requires athletes to again focus on maximal jump height, whilst also minimizing ground contact time (GCT) [7].

Typically, jump performance has been discussed pertaining to the jump height measurement during CMJ, DJ, and CMRJ [5,7,8]. However, athletes may use a variety of take-off strategies (i.e., hip or knee dominant), resulting in different levels of joint work during the propulsive phase, whilst still achieving the same jump height [9]. Meanwhile, simply measuring jump height alone is a poor reflection of the utilization of SSC mechanics to store and use elastic strain energy [10]. Therefore, joint kinetic analysis focusing on the total positive or negative joint work (and total joint contribution) offers a more detailed insight into how an athlete utilizes their lower extremity joints (i.e., hip, knee, and ankle) to achieve a given outcome [10,11]. For example, Hubley and Wells [11] conducted research to see the primary joint contributor to the total joint work done during a CMJ. They revealed that the knee joint (mean work: 330 ± 111 joule [J]) contributed to around 49% of the total positive work, with the remaining 51% of the work done by the ankle (161 ± 51 J; 23%) and hip joints (188 ± 76 J; 28%). In support of this, Lee et al. [12] argued that the work done by the ankle (2.06 J/kg) and knee joints (1.94 J/kg) were similar during a CMJ, and these values also remained unchanged when jump height increased. However, they also revealed that the hip joint work showed a significant increase from 1.03 J/kg to 3.24 J/kg as jump height increased ($p \leq 0.001$, omega squared $w^2 = 0.79$), highlighting the biggest contributor to total positive work was actually the hip joint. More recently, Harry et al. [10] proposed that all three lower extremity joints contributed to different positive and negative work (i.e., a net extensor moment combined with joint extension is presented as positive, a net flexor moment combined with joint flexion is negative) during the CMJ. In accordance with Hubley and Wells [11], Harry et al. [10] found the knee joint contributed to the majority of total negative and positive works (negative: 54.52 ± 14.09%; positive: 41.43 ± 8.26%) compared to the hip (negative: 39.78 ± 15.17%; positive: 32.14 ± 8.68%) and ankle joints (negative: 5.71 ± 3.69%; positive: 26.44 ± 4.85%). A couple of variables including the involvement of arm swing, the countermovement depth (CM depth), a variety of take-off strategies, or the sex of participants might explain the differences seen from the aforementioned studies [9,10,12].

When considering the DJ, two previous studies have found that the hip joint only contributed 13% and 19% of total positive work done [13,14]. These values were notably less than the CMJ (23% to 39%) [11,14], pointing to the notion that the ankle and knee joints contributed a greater portion to the total positive work done in the DJ. It is not surprising to see the primary contributor to the total positive work done is different between the CMJ and DJ, given that these two jump actions focus on different SSC mechanics [4,15]. Holcomb et al. [15] revealed that the total joint work was determined by the range of motion of lower extremity joints during DJ. Specifically, the joint that has the largest degree of flexion attenuates the majority of the landing force and thereby generating significantly higher negative works, compared to other joints where flexion is minimized [15]. In addition, Decker and McCaw [16] reported that the ankle joint was the primary contributor (45–50%), with the knee and hip joints being the second (30–35%) and third (15–25%) contributors to the total negative work during 60 cm drop heights (target heights were manipulated to 40, 60, and 80 cm). Similar results were supported by Yoshida et al. [17], whereby the joint work were largest in the ankle joint (negative: ~−1.00 to −2.20 J/kg; positive: ~1.60 to 2.00 J/kg), followed by the knee (negative: ~−0.75 to −1.60 J/kg; positive: ~0.85 to 1.00 J/kg) and hip joints (negative: ~−0.20 to −0.50 J/kg; positive: ~0.50 to 0.85 J/kg) at all drop heights (30, 60, 90, 120 cm). The aforementioned results partly support the notion that the ankle and knee joints are likely to be the biggest contributors to overall joint work done during the DJ [15]. Thus, this raises the question as to whether athletes will still rely on the ankle and knee joints during the rebound jump during the CMRJ, given its similarities.

There is currently a scarcity of studies investigating joint work during the CMRJ via a comprehensive joint kinetic analysis. For instance, Kariyama et al. [18] compared jump heights between single and double leg CMRJ, showing that the mean differences in hip joint works in the sagittal (negative: ~$-0.56$ J/kg; positive: ~0.10 J/kg) and frontal planes (negative: ~$-0.75$ J/kg; positive: ~0.95 J/kg) during the instant at take-off largely explained the discrepancy in jump heights. However, to the best of our knowledge, the difference in the total works of lower extremity joints during the braking and propulsive phases of CMRJ and the reliability of measured metrics in CMRJ have yet to be investigated. Therefore, the aims of this study were to (1) investigate the between-session reliability of metrics measured during CMRJ, CMJ, and DJ tests and (2) analyze the jump strategy employed in these actions by comparing differences between kinetic variables (negative and positive joint work and joint contribution to total work), jump height, and strategy-based metrics measured from CMJ, DJ, and CMRJ tests. Understanding the reliability of metrics in these jump actions and the similarity between CMRJ, CMJ, and DJ tests will provide valuable insights for practitioners looking to utilize CMRJ as an alternative assessment method [9,11,15]. It was hypothesized that (1) metrics measured in the CMRJ would demonstrate similar reliability to the other two tests and (2) metrics measured between jumps would show nonsignificant differences [18].

## 2. Materials and Methods

### 2.1. Participants

G power analysis (G*Power software, v. 3.1.9.7) indicated at least 26 participants were needed to have 80% chance (power equals to 0.8) to find the statistically significant difference ($p < 0.05$) between groups with a moderate effect size (Hedges' $g = 0.5$) [19]. Thirty-three physically active sport science students (age: $27.2 \pm 5.9$ years, height: $1.78 \pm 0.8$ cm, body mass: $77.5 \pm 11.5$ kg) volunteered to participate in this study, with none being professional athletes beyond the collegiate level at the time of participation. All participants were free of any lower body musculoskeletal injuries in the preceding six months and had experience of strength and plyometric training for at least one year. Each subject signed a written informed consent form before the commencement of any testing. This study was approved by the London Sport Institute research and ethics committee at Middlesex University (Application No: 21808).

This study administered a test-retest design, aiming to compare the negative and positive joint work, the individual joint contribution to the total work, jump height, and strategy-based metrics (including time to take-off [TTTO], countermovement depth [CM depth], and ground contact time [GCT]) measured during the CMJ, DJ, and CMRJ tests [1]. Participants attended two testing sessions and completed three trials of unloaded CMJ, DJ, and CMRJ with maximal effort in both testing sessions. The first session started with demonstrations of three jumps, participants were given five minutes to familiarize themselves with each jump type after the warm-up was completed, and then the jump testing commenced. The second session mirrored the first session and was implemented 48–96 h later. All variables measured during the first and second test sessions were compared to aid in interpreting the reliability and systematic bias between three jump types in two sessions.

### 2.2. Procedures

The participants' body height, age, and sex were recorded first, and then a code was given to represent their names to comply with data protection policies at the university. Test order for a total of nine jumps was randomized in session one to minimize any potential fatigue or learning effects impacting one specific jump test, and the same test order was retained for each subject in test session two [20]. Ninety seconds (s) of rest was set between each trial to ensure adequate recovery and minimize any potential fatigue impacting any of the jump tests [20]. Participants had ten minutes to perform low-intensity jogging and five minutes to perform dynamic stretching, which included forward lunge rotations, 'spiderman' with thoracic rotation, and forward and lateral hip swings [20]. Before data

collection, a static calibration trial was recorded first for each subject, with participants keeping their hands on their hips and standing upright. The arm swing was inhibited in this study to avoid any influences of upper-limb movements on the center of mass (COM) location estimation [21]. Depending on the jump types, the specific verbal cues such as "jump as high and as fast as you can" was given for the CMJ and CMRJ1, and "jump as high as you can whilst spending the shortest time possible on the ground" was given for the DJ and CMRJ2. These external verbal cues direct participants' attention to the movement outcomes only and ensure they utilize self-preferred jump strategies. Furthermore, no comments regarding the jump performance or segment movements were given to participants during testing [22].

Participants completed all jumps on twin embeddable force plates ([FP], 9281EA, Kistler Instruments Ltd., Hook, UK) at a sampling frequency of 1000 Hz in conjunction with 11 Qualisys three-dimensional motion capture cameras (Qualisys Oqus, Göteborg, Sweden) recording at a sampling frequency of 200 Hz. The FP recorded the ground reaction force (GRF) and was synchronized with the cameras. Forty-two reflective markers were bilaterally placed on specific anatomical land markers, as shown in Figure 1. These markers were modified from previous studies that investigated the lower limb joint kinetics and kinematics during jumping [10,23]. The data acquisition involved use of Qualisys Track Manager software (QTM, Qualisys, version 2.16, Gothenburg, Sweden) to collect the GRF data from the FP and marker positions data from the cameras. The FP was zeroed each time before participants stepped on, and participants were required to maintain fully extension of their lower extremities during take-off and landing in all jump tests [21,24]. During the CMJ and CMRJ, participants initiated movements by standing on the FP and ended by landing at the same FP as take-off [21]. The DJ was measured via twin embeddable FP using one FP method as proposed by McMahon et al. [24]. The box height of 0.30 m was chosen in accordance with previous investigations [24,25], and the participants stepped off the drop box (0.30 m) and came in contact with the FP placed adjacent to the drop box and rebounded immediately. For an accurate calculation of body weight, participants were also required to keep the same static position identical to the calibration trial for at least one second before initiating the CMJ (and CMRJ) and after landing from the DJ [7,24].

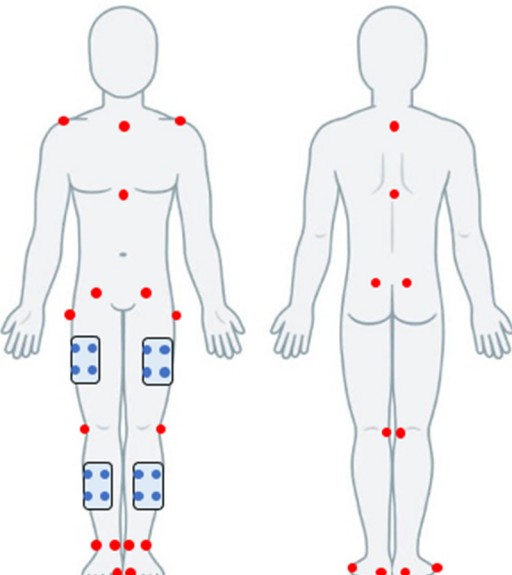

**Figure 1.** Anatomical body land marker locations of the 26 single reflective markers and four sets of non-collinear cluster markers were adhered symmetrically on both sides of each participants' body.

### 2.3. Data Processing

The marker data were digitized and labelled via QTM [15], and the recorded data were exported to Visual3D biomechanical software (v2023.09.3, C-Motion, Inc., Germantown, MD, USA) for the further computation of kinetic variables. A seven-segment rigid-line body model was built to represent the trunk, pelvis, left- and right-side thigh, shank, and foot segments. Then, this model was used to compute the motion of each segment and joint using a Cardan angle sequence x-y-z [10], where x represents the medial-lateral axis, y represents the anterior-posterior axis, and z represents the longitudinal axis. The raw GRF and marker data were smoothed via a low-pass Butterworth filter with 50 Hz and 12 Hz cut-off frequencies, respectively [10]. The sagittal plane joint power was calculated using Newtonian inverse dynamics, which is the product of the net joint moment and the corresponding joint angular velocity [10,12]. The GRF and joint power that had been processed in Visual3D were then exported to MATLAB (R2022a; The MathWorks, Inc., Natick, MA, USA) for further calculation.

The GRF was down-sampled five times to correspond to the motion data at 200 Hz [12] to calculate time-synchronized vertical velocity and displacement that acted on the participants' COM consistent with the previous literature [10,26]. The braking phase of the CMJ was defined as the time between the local peak negative COM velocity to the COM velocity reaching zero [26]. A time interval between the COM velocity reaching zero and the take-off instant (the net GRF being lower than five times the standard deviation of GRF extracted from only the middle part of the flight phase) was defined as the propulsive phase of the CMJ [26,27]. The touchdown of the CMJ was defined as the GRF value being larger than five times the standard deviation of GRF extracted from the flight phase [27]. The first touchdown during the DJ was identified as the GRF being larger than five times the standard deviation of vertical GRF extracted from the standing still period (i.e., participants stood on the drop box) [24]. Subsequently, the time between the local peak negative COM velocity and the local lowest COM vertical position was defined as the braking phase of the DJ [24]. The propulsive phase of the DJ was determined as the time between the local lowest COM vertical position and take-off instant (using the same force threshold as the CMJ) [24]. The method to define the braking and propulsive phases in the CMRJ1 was identical to CMJ, while the method to define these phases in CMRJ2 was identical to DJ [7]. Furthermore, the jump height of three jumps in the present study was calculated via the impulse-momentum methods from the COM height at the take-off instant to the height point of the flight phase [5]; the strategy-based metrics were calculated in accordance with methods described by Bishop et al. [1].

The mean negative and positive net joint works were calculated by integrating (through trapezoid rule) the negative and positive portions of the power–time curve during braking and propulsive phases of each jump action [10,16,28]. The negative work reflects energy absorption and positive work reflects energy generation [16]. The negative and positive hip, knee, and ankle joint work values were summed first to obtain the total negative and positive work, respectively [10]. Finally, the negative and positive work of each joint was divided by the total negative and positive work values to calculate the individual joint contribution to the total negative and positive lower extremity work [11]. All kinetic variables (joint work and contributions) were normalized to participants' body mass to reduce its influence on statistical comparison.

### 2.4. Statistical Analysis

The mean and standard deviation (SD) of all variables were recorded and taken forward for statistical analysis in SPSS (version 27; SPSS Inc., Chicago, IL, USA) and Microsoft Excel. Normality of data was assessed by the Shapiro–Wilk statistic, and homogeneity of variance was verified with Levene's test. A two-way repeated measures analysis of variance (ANOVA, session $\times$ jump type) was used to determine the differences in all measured metrics from three jump types between sessions, with statistical significance set at $p < 0.05$. The between-session reliability for all measured variables were calculated via a 2-way random

model intraclass correlation coefficients (ICC) with 95% confidence intervals (CI) and the coefficient of variation (CV) with 95% CI [29]. The magnitude of ICC was assessed based on suggestions by Koo and Li [30], with an ICC value > 0.90 = excellent, 0.75–0.90 = good, 0.50–0.74 = moderate, and <0.50 = poor. CV was calculated as (CV% = SD/mean × 100), with values considered good if <5%, moderate if between 5 and 10%, and poor if >10% [31]. The 95% CI for CV was calculated as $((CV/\sqrt{(2n)}) \times 1.96)$, where $n$ was the total numbers of participants. In addition, Hedges' $g$ effect sizes (ES) were calculated in Microsoft Excel to provide an explanation of practical significance between test sessions, as $((Mean_{session1} - Mean_{session2})/SD$ pooled) [32], where the ES values were interpreted as $g < 0.35$ = trivial; 0.35–0.80 = small; 0.81–1.50 = moderate; and >1.5 = large [33].

## 3. Results

### 3.1. Reliability

The between-session reliability results for the CMJ, DJ, and CMRJ are presented in Table 1. The CMJ showed good to excellent relative reliability and good to poor CV for almost all variables (ICC = 0.75–0.98; CV ≤ 20.95%), excluding the positive ankle contribution that showed only moderate reliability (ICC = 0.73). The reliability measured in CMRJ1 followed a similar trend to the CMJ, with all metric but negative hip work (ICC = 0.73) being ≥0.80 and deemed good to excellent reliability with moderate to poor CV (≤17.63%). Finally, apart from the hip joint related variables that showed moderate to good reliability with poor CV (ICC = 0.64–0.89; CV ≤ 25.04%), all other variables measured in DJ and CMRJ2 present good to excellent relative reliability with moderate CV (ICC = 0.82–0.97; CV ≤ 9.91%).

**Table 1.** Mean within-session data (±SD) and between-session reliability for performance metrics and individual joint work and contribution to the total works reported from the average of all trials.

| Test | Joint | Variables | Test Session 1 | Test Session 2 | Between-Session | | |
|---|---|---|---|---|---|---|---|
| | | | Mean ± SD | Mean ± SD | Hedges' $g$ (95% CI) | Descriptor | ICC (95% CI) | CV (95% CI) |
| CMJ | | CM Depth | 0.29 ± 0.05 | 0.29 ± 0.06 | −0.06 (−0.28, 0.16) | Trivial | 0.89 (0.78, 0.95) | 7.20 (5.38, 9.02) |
| | | TTTO | 0.74 ± 0.10 * | 0.74 ± 0.13 * | 0.00 (−0.27, 0.27) | Trivial | 0.82 (0.64, 0.91) | 6.34 (4.74, 7.95) |
| | | Jump Height | 0.31 ± 0.08 | 0.32 ± 0.08 | −0.05 (−0.16, 0.06) | Trivial | 0.98 (0.95, 0.99) | 4.19 (3.13, 5.24) |
| | Hip | Negative Work | −0.51 ± 0.21 [a] | −0.52 ± 0.17 [a] | 0.06 (−0.43, 0.55) | Trivial | 0.78 (0.55, 0.89) | 15.35 (11.64, 19.05) |
| | | Positive Work | 0.93 ± 0.47 [b] | 0.99 ± 0.43 [b] | −0.14 (−0.63, 0.36) | Trivial | 0.81 (0.61, 0.91) | 20.95 (15.90, 26.00) |
| | Knee | Negative Work | −0.91 ± 0.27 [c] | −0.99 ± 0.32 [c] | 0.25 (−0.25, 0.74) | Small | 0.89 (0.76, 0.95) | 8.83 (6.70, 10.96) |
| | | Positive Work | 2.51 ± 0.62 | 2.57 ± 0.72 | −0.09 (−0.58, 0.40) | Trivial | 0.88 (0.76, 0.94) | 8.50 (6.45, 10.55) |
| | Ankle | Negative Work | −0.16 ± 0.06 | −0.15 ± 0.07 | −0.08 (−0.57, 0.42) | Trivial | 0.89 (0.78, 0.95) | 15.35 (11.64, 19.05) |
| | | Positive Work | 1.86 ± 0.62 | 1.85 ± 0.69 | 0.04 (−0.46, 0.53) | Trivial | 0.86 (0.71, 0.93) | 5.84 (4.43, 7.25) |
| | Hip | Negative Contribution | 30.87 ± 10.75 | 31.76 ± 10.25 | −0.08 (−0.58, 0.41) | Trivial | 0.87 (0.74, 0.94) | 13.03 (9.89, 16.17) |
| | | Positive Contribution | 17.04 ± 7.45 | 18.12 ± 7.15 | −0.15 (−0.64, 0.35) | Trivial | 0.76 (0.52, 0.88) | 19.29 (14.64, 23.94) |
| | Knee | Negative Contribution | 58.02 ± 10.25 | 58.87 ± 9.98 | −0.08 (−0.58, 0.41) | Trivial | 0.85 (0.70, 0.93) | 6.74 (5.11, 8.36) |
| | | Positive Contribution | 47.41 ± 7.21 | 47.29 ± 8.04 | 0.02 (−0.48, 0.51) | Trivial | 0.75 (0.50, 0.88) | 5.59 (4.24, 6.93) |
| | Ankle | Negative Contribution | 10.08 ± 3.91 | 9.36 ± 3.87 | 0.18 (−0.31, 0.68) | Trivial | 0.87 (0.73, 0.93) | 15.57 (11.81, 19.33) |
| | | Positive Contribution | 35.54 ± 4.18 | 35.12 ± 4.59 | −0.15 (−0.64, 0.35) | Trivial | 0.73 (0.44, 0.86) | 3.89 (2.95, 4.82) |
| CMRJ1 | | CM Depth | 0.27 ± 0.05 | 0.27 ± 0.06 | 0.05 (−0.15, 0.26) | Trivial | 0.91 (0.81, 0.95) | 7.01 (5.24, 8.78) |
| | | TTTO | 0.67 ± 0.11 * | 0.68 ± 0.13 * | −0.02 (−0.31, 0.27) | Trivial | 0.80 (0.59, 0.90) | 8.64 (6.45, 10.83) |
| | | Jump Height | 0.30 ± 0.07 | 0.29 ± 0.07 | 0.07 (−0.05, 0.20) | Trivial | 0.97 (0.93, 0.98) | 4.86 (3.63, 6.09) |
| | Hip | Negative Work | −0.43 ± 0.15 [a] | −0.42 ± 0.15 [a] | −0.05 (−0.54, 0.44) | Trivial | 0.74 (0.46, 0.87) | 17.63 (13.38, 21.88) |
| | | Positive Work | 0.78 ± 0.42 [b] | 0.76 ± 0.41 [b] | 0.06 (−0.44, 0.55) | Trivial | 0.90 (0.79, 0.95) | 16.69 (12.59, 20.59) |
| | Knee | Negative Work | −0.77 ± 0.20 [c] | −0.81 ± 0.25 [c] | 0.17 (−0.32, 0.66) | trivial | 0.91 (0.82, 0.96) | 8.65 (6.56, 10.73) |
| | | Positive Work | 2.41 ± 0.61 | 2.33 ± 0.68 | 0.12 (−0.37, 0.62) | Trivial | 0.90 (0.79, 0.95) | 7.92 (6.01, 9.59) |
| | Ankle | Negative Work | −0.14 ± 0.06 | −0.14 ± 0.07 | 0.00 (−0.49, 0.49) | Trivial | 0.87 (0.73, 0.93) | 15.58 (11.82, 19.34) |
| | | Positive Work | 1.81 ± 0.30 | 1.77 ± 0.34 | 0.13 (−0.37, 0.62) | Trivial | 0.93 (0.85, 0.96) | 5.26 (3.99, 6.53) |
| | Hip | Negative Contribution | 31.52 ± 9.21 | 30.42 ± 9.96 | 0.11 (−0.38, 0.60) | Trivial | 0.86 (0.71, 0.93) | 11.64 (8.83, 15.45) |
| | | Positive Contribution | 15.15 ± 7.48 | 15.35 ± 7.86 | −0.03 (−0.52, 0.47) | Trivial | 0.87 (0.73, 0.93) | 13.29 (10.08, 16.49) |
| | Knee | Negative Contribution | 57.85 ± 9.41 | 59.23 ± 9.76 | −0.14 (−0.63, 0.35) | Trivial | 0.87 (0.74, 0.94) | 5.00 (3.80, 6.21) |
| | | Positive Contribution | 48.14 ± 7.44 | 47.84 ± 8.03 | 0.04 (−0.45, 0.53) | Trivial | 0.85 (0.70, 0.93) | 3.86 (2.93, 4.79) |
| | Ankle | Negative Contribution | 10.63 ± 3.68 | 10.35 ± 3.85 | 0.07 (−0.42, 0.57) | Trivial | 0.87 (0.73, 0.93) | 14.43 (10.95, 17.91) |
| | | Positive Contribution | 36.13 ± 5.04 | 36.24 ± 5.07 | −0.02 (−0.51, 0.47) | Trivial | 0.88 (0.75, 0.94) | 3.75 (2.84, 4.65) |

**Table 1.** *Cont.*

| Test | Joint | Variables | Test Session 1 | Test Session 2 | Between-Session | | |
|------|-------|-----------|------|------|------|------|------|
| | | | Mean ± SD | Mean ± SD | Hedges' *g* (95% CI) | Descriptor | ICC (95% CI) | CV (95% CI) |
| DJ | | GCT | 0.32 ± 0.11 | 0.32 ± 0.11 | 0.05 (−0.07, 0.17) | Trivial | 0.97 (0.94, 0.99) | 5.91 (4.42, 7.41) |
| | | Jump Height | 0.28 ± 0.07 | 0.27 ± 0.07 | 0.03 (−0.13, 0.19) | Trivial | 0.95 (0.89, 0.97) | 5.97 (4.46, 7.47) |
| | Hip | Negative Work | −0.67 ± 0.28 | −0.66 ± 0.32 | −0.05 (−0.54, 0.44) | Trivial | 0.73 (0.44, 0.87) | 20.01 (15.18, 24.83) |
| | | Positive Work | 0.61 ± 0.50 | 0.60 ± 0.42 | 0.03 (−0.46, 0.52) | Trivial | 0.88 (0.76, 0.94) | 25.04 (19.00, 31.09) |
| | Knee | Negative Work | −1.97 ± 0.64 [d] | −1.87 ± 0.74 [d] | −0.14 (−0.63, 0.35) | Trivial | 0.95 (0.89, 0.97) | 9.35 (7.86, 11.85) |
| | | Positive Work | 2.06 ± 0.60 [e] | 2.02 ± 0.72 [e] | 0.06 (−0.43, 0.55) | Trivial | 0.91 (0.82, 0.96) | 9.91 (7.52, 12.30) |
| | Ankle | Negative Work | −1.47 ± 0.41 | −1.50 ± 0.52 | 0.06 (−0.43, 0.55) | Trivial | 0.93 (0.85, 0.96) | 8.46 (6.42, 10.50) |
| | | Positive Work | 1.93 ± 0.29 | 1.92 ± 0.33 | 0.02 (−0.47, 0.51) | Trivial | 0.88 (0.75, 0.94) | 5.81 (4.41, 7.21) |
| | Hip | Negative Contribution | 16.17 ± 5.57 | 16.01 ± 6.27 | 0.03 (−0.47, 0.52) | Trivial | 0.75 (0.49, 0.88) | 17.80 (13.50, 22.09) |
| | | Positive Contribution | 12.14 ± 8.00 | 12.43 ± 7.77 | −0.04 (−0.53, 0.46) | Trivial | 0.88 (0.75, 0.94) | 21.99 (16.69, 27.30) |
| | Knee | Negative Contribution | 47.01 ± 10.92 | 45.66 ± 13.38 | 0.11 (−0.38, 0.60) | Trivial | 0.93 (0.85, 0.96) | 8.64 (6.55, 10.72) |
| | | Positive Contribution | 44.46 ± 7.22 | 43.34 ± 8.74 | 0.14 (−0.35, 0.63) | Trivial | 0.85 (0.70, 0.93) | 6.76 (5.13, 8.39) |
| | Ankle | Negative Contribution | 36.82 ± 12.50 | 38.33 ± 15.07 | −0.11 (−0.60, 0.38) | Trivial | 0.93 (0.86, 0.97) | 8.96 (6.80, 11.12) |
| | | Positive Contribution | 43.40 ± 8.70 [f] | 44.24 ± 10.42 [f] | −0.09 (−0.58, 0.41) | Trivial | 0.82 (0.63, 0.91) | 5.56 (4.22, 6.95) |
| CMRJ2 | | GCT | 0.35 ± 0.12 | 0.34 ± 0.11 | 0.04 (−0.12, 0.20) | Trivial | 0.94 (0.89, 0.97) | 8.13 (6.07, 10.19) |
| | | Jump Height | 0.28 ± 0.07 | 0.28 ± 0.07 | −0.02 (−0.20, 0.16) | Trivial | 0.93 (0.87, 0.97) | 6.08 (4.54, 7.62) |
| | Hip | Negative Work | −0.82 ± 0.33 | −0.76 ± 0.44 | −0.14 (−0.64, 0.35) | Trivial | 0.64 (0.28, 0.82) | 18.94 (14.37, 23.51) |
| | | Positive Work | 0.70 ± 0.52 | 0.71 ± 0.46 | −0.01 (−0.50, 0.49) | Trivial | 0.89 (0.77, 0.94) | 21.49 (16.31, 26.68) |
| | Knee | Negative Work | −2.51 ± 0.84 [d] | −2.41 ± 0.84 [d] | −0.12 (−0.61, 0.38) | Trivial | 0.90 (0.86, 0.94) | 9.74 (7.91, 11.58) |
| | | Positive Work | 2.41 ± 0.67 [e] | 2.30 ± 0.68 [e] | 0.16 (−0.33, 0.65) | Trivial | 0.91 (0.87, 0.97) | 9.22 (7.03, 11.50) |
| | Ankle | Negative Work | −1.47 ± 0.46 | −1.54 ± 0.52 | 0.15 (−0.35, 0.64) | Trivial | 0.82 (0.63, 0.91) | 9.03 (6.85, 11.21) |
| | | Positive Work | 1.90 ± 0.31 | 1.90 ± 0.34 | 0.00 (−0.49, 0.49) | Trivial | 0.90 (0.88, 0.96) | 6.18 (4.69, 7.67) |
| | Hip | Negative Contribution | 16.54 ± 5.72 | 15.81 ± 7.27 | 0.11 (−0.38, 0.60) | Trivial | 0.85 (0.72, 0.94) | 17.24 (13.08, 21.40) |
| | | Positive Contribution | 12.77 ± 8.13 | 13.65 ± 8.45 | −0.11 (−0.60, 0.39) | Trivial | 0.89 (0.77, 0.94) | 24.11 (18.30, 29.93) |
| | Knee | Negative Contribution | 51.49 ± 11.70 | 50.07 ± 11.81 | 0.12 (−0.37, 0.61) | Trivial | 0.82 (0.63, 0.91) | 6.35 (4.82, 7.89) |
| | | Positive Contribution | 47.58 ± 7.96 | 46.33 ± 7.58 | 0.16 (−0.33, 0.65) | Trivial | 0.86 (0.72, 0.93) | 5.51 (4.18, 6.84) |
| | Ankle | Negative Contribution | 31.91 ± 12.15 | 33.82 ± 13.44 | −0.15 (−0.64, 0.35) | Trivial | 0.85 (0.69, 0.92) | 7.23 (5.49, 8.97) |
| | | Positive Contribution | 38.70 ± 8.92 [f] | 40.02 ± 9.01 [f] | −0.15 (−0.64, 0.35) | Trivial | 0.83 (0.66, 0.92) | 8.82 (6.69, 10.95) |

[*] Significant different TTTO between CMJ and CMRJ1 in both test sessions 1 ($p < 0.001$) and 2 ($p < 0.001$). [a] Significant different negative hip work between CMJ and CMRJ1 in both test sessions 1 ($p = 0.005$) and 2 ($p = 0.019$). [b] Significant different positive hip work between CMJ and CMRJ1 in both test sessions 1 ($p = 0.026$) and 2 ($p = 0.026$). [c] Significant different negative knee work between CMJ and CMRJ1 in both test sessions 1 ($p = 0.035$) and 2 ($p = 0.009$). [d] Significant different negative knee work between DJ and CMRJ2 in both test sessions 1 ($p = 0.005$) and 2 ($p = 0.005$). [e] Significant different positive knee work between DJ and CMRJ2 in both test sessions 1 ($p = 0.034$) and 2 ($p = 0.020$). [f] Significant different positive ankle contribution between DJ and CMRJ2 in both test sessions 1 ($p = 0.048$) and 2 ($p = 0.042$). Units of measure for coefficient of variation: %; Units of measure for total negative and positive works: Joules per kilogram system mass (J/kg); Unit of measure for hip, knee, and ankle contributions: percentage of total positive/negative work (% Total); Unit of measure for CM depth and jump height: meter; Unit of measure for GCT: second. CMJ = countermovement jump; CMRJ1 = the first jump of the countermovement rebound jump; DJ = drop jump; CMRJ2 = the second jump of the countermovement rebound jump; CM depth: countermovement depth; TTTO: time to take-off; GCT: ground contact time.

### 3.2. Comparison between Jump Tests

Results from ANOVA showed no significant differences across all metrics were present between two test sessions during CMJ ($p \geq 0.129$), DJ ($p \geq 0.079$), and CMRJ ($p \geq 0.161$) with the ES values presented in Table 1. The jump height was not significantly different between CMJ and CMRJ1 ($p \geq 0.118$) and DJ and CMRJ2 ($p \geq 0.812$) during both test sessions 1 and 2. For the strategy-based metrics, the TTTO from the CMJ was significantly longer than CMRJ1 by 0.07 s in test session 1 ($p = 0.022$, $g = 0.62$) and 0.06 s in session 2 ($p = 0.027$, $g = 0.49$). Moreover, CM depth from the CMJ was not significantly deeper by 0.02 m in both sessions 1 ($p = 0.304$, $g = 0.27$) and 2 ($p = 0.192$, $g = 0.32$) compared to the CMRJ1. The nonsignificant small differences in the GCT between the DJ and CMRJ2 were observed in both sessions 1 ($p \leq 0.308$, $g = -0.24$) and 2 ($p \leq 0.324$, $g = -0.24$).

The comparisons of kinetic variables from different jump actions were also presented in Table 1, and Figures 2–4. In both test sessions, the negative hip ($p \leq 0.019$, $g \leq 0.62$) and knee ($p \leq 0.035$, $g \leq 0.58$) joint works and positive hip joint works ($p \leq 0.029$, $g \leq 0.54$) were found to be significantly larger in the CMJ than CMRJ1. No significant differences were revealed in other joint work and individual joint contributions between CMJ and CMRJ1 ($p \geq 0.137$). Compared to the CMRJ2, the positive ($p \leq 0.034$, $g \leq 0.54$) and negative knee work ($p = 0.005$, $g \leq 0.71$) from the DJ were significantly smaller in both sessions. A significantly larger ankle contribution to the total positive work was observed in DJ than CMRJ2 in both sessions 1 ($p = 0.048$, $g = 0.53$) and 2 ($p = 0.042$, $g = 0.43$). Finally, no

significant differences were observed in other kinetic variables measured between DJ and CMRJ2 ($p \geq 0.086$).

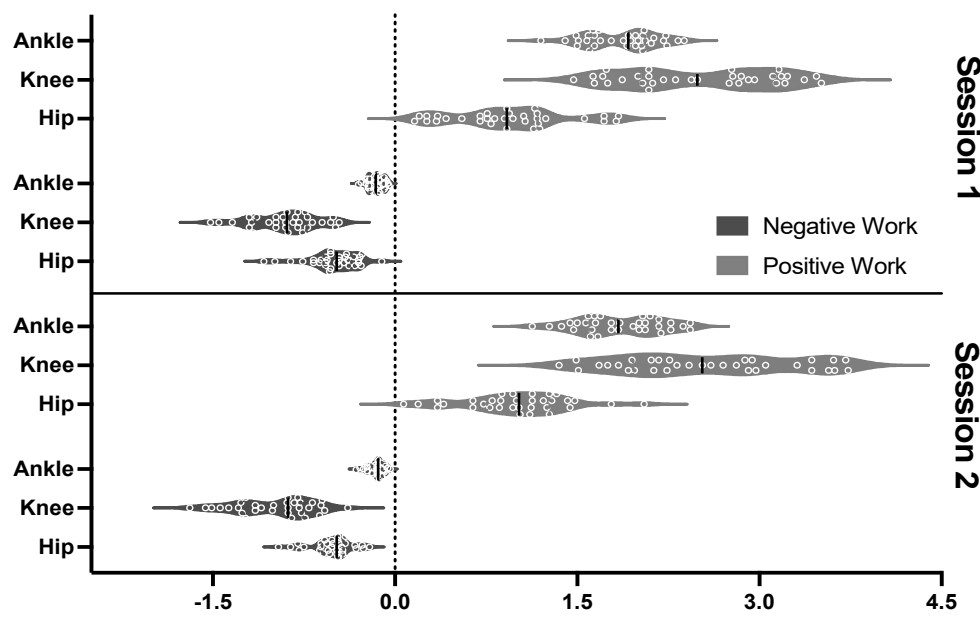

**Figure 2.** A summary of the individual joint contribution to the total work during three jump actions in both test sessions. The positive values indicate the contribution to the total positive work. The negative values indicate the contribution to the total negative work. The percentage numbers on the figure were rounded.

**Figure 3.** *Cont.*

## Countermovement Rebound Jump 1

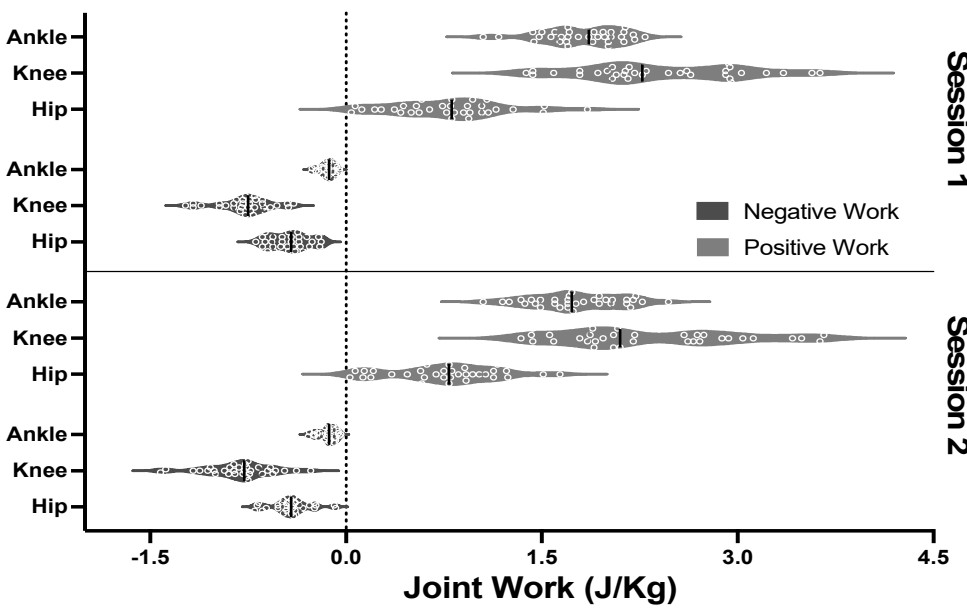

**Figure 3.** A summary of the total negative and positive work in each individual joint during countermovement jump and countermovement rebound jump 1 in both test sessions. The positive values indicate the positive work. The negative values indicate the negative work.

## Drop Jump

**Figure 4.** *Cont.*

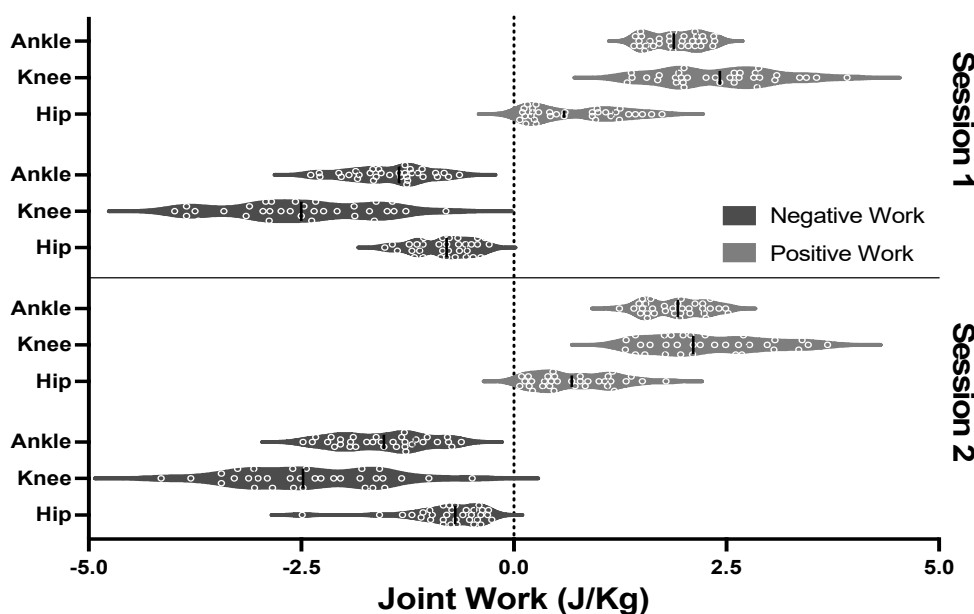

**Figure 4.** A summary of the total negative and positive work in each individual joint during drop jump and countermovement rebound jump 2 in both test sessions. The positive values indicate the positive work. The negative values indicate the negative work.

## 4. Discussion

This study aimed to (1) determine the between-session reliability of all metrics measured during the CMJ, DJ, and CMRJ tests and (2) determine the differences in measured variables between the three jump types during two testing sessions. The main findings of this study showed that all variables consistently demonstrated moderate to excellent reliability across all three jump types, while the negative and positive hip joint works and contributions in all three jump types and negative ankle work and contributions in CMJ and CMRJ1 showed good to poor CV. Of all strategy-based metrics, only TTTO revealed significant small differences between CMRJ against CMJ and DJ tests. Furthermore, some of the kinetic variables (negative and positive hip and knee works and positive ankle contribution) significantly differed between the jump tests with trivial to moderate ES.

In line with the previous research [7], the present study found that jump height and all strategy-based metrics exhibited good to excellent between-session reliability in each jump (ICC $\geq 0.89$), with good to moderate CV ($\leq 8.64\%$). Thus, our first hypothesis regarding the reliability of measured metrics was confirmed. In CMJ and CMRJ1, we observed questionable reliability in negative and positive hip joint work (CV $\leq 20.95\%$) and contributions (CV $\leq 19.29\%$), and negative ankle work (CV $\leq 15.58\%$) and contributions (CV $\leq 15.51\%$). This suggests an altered joint dominant strategy was used to achieve certain braking or propulsive impulse based on environmental factors, movement constraints, or other individual confounding factors [34]. Although speculative, the most influential factor may have been the unrestricted CM depth which enabled participants to manipulate their COM positions to a variety of heights between test sessions (as reflected by the moderate CV values of this metric). Naturally, the previous research has shown that variations in CM depth causes changes in joint angles during the braking and propulsive phases of the CMJ [35,36]. Considering that joint work in the present investigation was obtained by integrating the joint power, which is the product of joint force and angular velocity [10,12], it becomes evident that the variability in joint angles can have a substantial impact on the calculated joint work and contribution. In addition, the only variables in DJ and CMRJ2 showing questionable reliability were negative and positive hip work (CV $\leq 25.04\%$) and contributions (CV $\leq 24.11\%$). As aforementioned, no internal verbal cues were given to

direct participants' attention to segment movement patterns [22]. Since quickness and higher displacement were the only requirements to complete DJ and CMRJ2, participants relied primarily on their knee and ankle joints during the braking and propulsive phases. With this in mind, only a small portion of the total negative and positive work was done by the hip joint [16], which might in part explain the greater bandwidth of variations associated with the hip joint. Overall, variables such as the individual joint work and contribution to the total work appear to be more variable when monitoring jump performance. Practitioners, therefore, should be mindful of this when aiming to establish meaningful changes between testing sessions.

Systematic bias was assessed between test sessions for all metrics, with no significant differences reported. Considering the joint work and contributions during CMJ and CMRJ1 tests, the knee joint was the biggest contributor to the total negative and positive work, which was consistent with the previous studies [10,11,14,37]. In terms of the second and third contributors, Harry et al. [10] and Hubley and Wells [11] argued that they observed a slightly higher percentage of hip contribution ($32.14 \pm 8.68\%$ and 28%) than ankle contribution ($26.44 \pm 4.85\%$ and 23%) to the total positive work. In contrast, the present study found that the ankle joint contributed to ~36% of the total positive work, which is higher than that of the hip joint (~17%) in both CMJ and CMRJ1. The discrepancy between the current investigation and the previous literature is likely because the arm swing was restricted, where participants performed all jumps with an upright trunk position. That said, vertical jumps with arm swing have been shown to induce greater trunk extension from forward inclination and exhibit greater angular velocity during hip joint extension in the propulsive phase, ultimately increasing the positive hip contribution [12,37].

Regarding the variables between jumps, the results partially support our second hypothesis, where significantly higher negative hip and knee joint work ($p \leq 0.035, g \leq 0.62$) were observed in CMJ compared to CMRJ1 (Figure 3). It seems participants utilized different jump strategies during the CMJ via slightly deeper CM depth ($p \leq 0.304, g \leq 0.32$) and significantly longer TTTO ($p \leq 0.027, g \leq 0.62$) to maximize their jump heights. It has previously been suggested that initiating the propulsive phase from different knee flexion angles (i.e., different CM depth) would generate significantly different positive kinetic variables [38]. In further support of this, this study found significantly higher positive work in hip joint in the CMJ ($p \leq 0.029, g \leq 0.54$) during both sessions. Noting that although some of the joint work or strategy-based metrics were significantly different between CMJ and CMRJ1, jump height and individual joint contribution to the total work were not ($p \geq 0.115, g \leq 0.37$), and the knee joint still acted as the primary contributor across both jumps. Thus, when arm swing was inhibited during our methods, participants consistently adopted knee and hip dominant strategies to attenuate the negative work while adopting knee and ankle dominant strategies to generate the positive work (see Figure 2).

When comparing the DJ and CMRJ2, our results aligned with the previous studies [13,15,23], where the knee joint contributed the biggest amount of total negative and positive work, followed by the ankle and hip joints during both tests, as shown in Figure 2. However, authors in two studies argued that the biggest contributor to the total work was the ankle joint (45% to 51%), followed by the knee (30% to 37%) and hip (11% to 20%) joints [14,17]. A possible reason is that participants in this study were physically active students who lacked extensive jump training experience, while the previous two investigations recruited male handball players [14] and male jumpers [17] who were likely to be more familiar with fast SSC-based plyometrics. Additionally, the GCT values measured in the DJ and CMRJ2 were longer than 0.25 s, making it hard to suggest that fast SSC mechanics were truly evident. Therefore, it is unlikely that participants with minimal jump-specific training employed the same jump strategies or joint dominance as those who are more familiar with the task [16], which likely explains why contradictions appeared between our findings and the previous literature.

In terms of the variables between jump types, significantly higher negative knee work was observed in CMRJ2 by ~0.54 J/kg ($p = 0.005, g \leq 0.71$) in session 1 and ~0.54 J/kg

($p$ = 0.005, $g \leq 0.64$) in session 2 (Figure 4). This might be because the actual drop heights differed between the two jumps, where the DJ had a fixed drop height of 0.30 m, and the CMRJ2 utilized the jump height in CMRJ1 as the drop height, which occasionally exceeded 0.30 m. Yoshida et al. [17] partly supported this explanation, as they observed an increase in the negative work in all three joints after landing from higher heights (from 0.30 to 1.20 m). Herein, landing from greater heights necessitates forceful eccentric muscle actions of the lower limb muscles to attenuate the larger inertial force impacted on the body [13,14,16,17]. The significantly larger negative knee joint work in CMRJ2 suggested a potential increase in elastic strain energy stored in the tendons, which in turn may have contributed to a significantly larger positive knee joint work by ~0.40 J/kg compared to DJ ($p \leq 0.034$, $g \leq 0.54$) [10,23]. This finding aligns closely with Decker and McCaw [16], who reported higher positive work in all three joints when landing from 0.60 m than from 0.40 m. No significant differences in jump height were observed between the DJ and CMRJ2 ($p \geq 0.812$), whilst the GCT was slightly longer in CMRJ2 than in DJ by 0.03 s ($p \leq 0.324$, $g \leq -0.24$). A large contribution of the ankle joint to the total positive work has been shown to be associated with a shorter GCT [13,39]. Similarly, in our study, the ankle joint contributed to 43% of the total positive work in DJ, and this value was significantly lower in CMRJ2 ($p \leq 0.048$, $g \leq 0.53$). These disparities in GCT and ankle joint contribution between DJ and CMRJ2 suggest that participants may have used their ankle joints differently due to variations in drop heights and eccentric loads between the two jump types. Thus, our second hypothesis was partially confirmed, where the CMRJ2 test largely mimics the characteristics of the DJ test and provides practitioners with similar joint work and joint contribution percentages.

Differences in the jump strategies might result in variations in the joint work and joint contribution to the total work performed between the three jump tests. Concurrently focusing on more than two verbal cues during the CMRJ may add more challenge to participants than during the CMJ and DJ tests, and potentially may have been a greater learning effect for CMRJ compared to the other two tests. Furthermore, no familiarization session was involved in this study; instead, participants had only five minutes to familiarize themselves with the CMRJ in each session. Therefore, it seems plausible to suggest that as challenges increased (in the CMRJ), participants were likely to utilize a range of different jump strategies to achieve comparable movement outcomes to the CMJ and DJ tests. Given that only some metrics show significant differences across the three different jump types, and the contributions of the lower limb joints to total negative and positive work remained consistent between three jump actions (see Figure 2). Therefore, practitioners can confidently utilize the CMRJ as a viable alternative to CMJ and DJ tests, with the exception of the hip joint work and contributions that showed questionable reliability (CV $\leq$ 20.95%).

Some limitations of this study should be acknowledged. Firstly, the inverse dynamic method cannot determine individual muscle activity. For instance, if a relatively large positive knee joint work was found during jump actions, it is unknown whether this increase occurs due to the increased activity of the knee extensor or knee flexor muscles. Secondly, large SD values for the measured variables indicated the variability of foot stiffness and jump performance amongst the participants. In cases where participants exhibited weak foot stiffness, it was likely that kinetic energy had been lost as heat, leading to an underestimation of the calculated ankle joint work [23]. Consequently, this inter-subject variation indicated that the strategies at the ankle joint are abundant and more varied from trial to trial and ultimately result in different percentages of the ankle joint contributions to the total work performed compared to the previous literature [14,17]. Finally, when we consider the sporting background and training history of the present sample, future research may wish to consider separating participants into groups (e.g., high vs. low jumpers) based on their performance scores [28], enabling a more detailed understanding of how kinetic variables contribute to overall jump performance.

## 5. Conclusions

The findings of this research indicate two primary take-home messages. First, the ankle and knee joints contributed most to the total positive work, especially in DJ and CMRJ2. Therefore, to optimize the jump performance during the CMJ, DJ, and CMRJ tests, training emphasis should be to improve strength surrounding the knee and ankle joints. One approach to achieve this is through unloaded jump training, where participants deliberately minimize variations in hip joint angle by avoiding trunk forward inclination. Specifically, they should priorities minimizing GCT, which is likely to encourage reduced joint displacement and maximize work done at the knee and ankle joints. In addition, and to provide a balanced approach to training suggestions, given that nearly half of the negative joint work was absorbed by the knee joint, it is possible that excessive repetitive stress may cause an overuse injury to the knee joint [23]. Thus, training programs should also emphasize eccentric strength training surrounding the knee joint and the deceleration ability of the lower extremities while reinforcing proper landing mechanics.

**Author Contributions:** Conceptualization, J.X., A.T. and C.B.; methodology, J.X., A.T. and C.B.; software, J.X., A.T. and C.B.; validation, J.X., A.T. and C.B.; formal analysis, J.X., C.B. and S.C.; investigation, J.X., C.B. and S.C.; resources, J.X., A.T. and C.B.; data curation, J.X. and C.B.; writing—original draft preparation, J.X. and C.B.; writing—review and editing, J.X., A.T., S.C., T.M.C., J.R.H. and C.B.; visualization, J.X. and C.B.; supervision, A.T. and C.B.; project administration, A.T. and C.B. All authors have read and agreed to the published version of the manuscript.

**Funding:** This research received no external funding.

**Institutional Review Board Statement:** The study was conducted according to the guidelines of the Declaration of Helsinki, according to current national laws and regulations, and approved by the by the London Sport Institute research and ethics committee at Middlesex University (Application No: 21808).

**Informed Consent Statement:** Informed consent was obtained from all subjects involved in the study.

**Data Availability Statement:** The raw data supporting the conclusions of this article will be made available by the authors without undue reservation.

**Acknowledgments:** The authors warmly thank the participants of this study.

**Conflicts of Interest:** The authors declare no conflict of interest.

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
