# Peer review of "Countermovement Rebound Jump: A Comparison of Joint Work and Joint Contribution to the Countermovement and Drop Jump Tests"

_applsci, doi:10.3390/app131910680_

Round 1

Reviewer 1 Report

The authors investigated the kinematic and kinetic differences between countermovement, drop, and countermovement rebound jumps performed by a large sample of physically active men. The authors also reported the inter-session reliability for the mechanical variables measured. Importantly, the authors conducted an analysis of the positive and negative work performed by the moments at the hip, knee, and ankle joints. The authors have presented information that practitioners will find useful and the manuscript is largely well-written, although there are numerous grammatical errors throughout that require revision and the authors need to ensure that their methods and procedures are clear. Therefore, I am recommending that the authors revise the manuscript based upon the following:

Introduction, line 54: Change “...joint works...” to “...joint work...”

Introduction, line 67: Change “works” to “work”. Also, note here that an extensor moment is presented as positive by convention, but an extensor moment only does positive work if the motion of the joint is also positive (i.e. a net extensor moment combined with extension at the joint). Consider altering the note presented in the parentheses here.

Introduction, line 72: Change “A couple of reasons...” to “Variables including the involvement of an arm swing, the countermovement depth...”

Introduction, line 84: I do not think that you need to include the alpha used in the Holcomb et al. study here as it adds very little. An effect size would be more useful.

Introduction, lines 85-88: You cite the work of Decker and McCaw here, noting that the authors assessed joint work during “40, 60, and 80 cm drop heights.” However, these authors actually had participants drop from a constant height of 60 cm; it was actually the target heights that were manipulated to 40, 60, and 80 cm. Please correct this.

Introduction, line 88: Change “works” to “work”.

Introduction, line 107: Change “works” to “work” (twice).

Materials and Methods, line 128: Change “works” to “work” (twice).

Materials and Methods, line 130: Present a reference for “strategy-based metrics”. Perhaps Bishop et al. (2021).

Materials and Methods, line 159: Present the model number of the force plates used.

Materials and Methods, line 172: You note that a single force plate was used for the DJ. If this is the case, then the calculation of bilateral joint kinetics would not be possible as each leg requires the specific GRF on that side of the body to be measured. Please correct this.

Materials and Methods, line 173: Please present the rationale for using a 30 cm box.

Materials and Methods, line 193: Presumably joint angular velocities were calculated using first central difference methods? Please add this.

Materials and Methods, line 198: Remove the parenthesis after COM here.

Materials and Methods, line 201: You note “the net GRF being lower than five times the net GRF extracted from only the middle part of the flight phase.” Do you mean 5 x the standard deviation of the GRF during flight, or actually 5 x the GRF during flight? See also lines 203-204 and line 205.

Materials and Methods, lines 206-208: The braking phase here includes the “yielding” phase that is absent from your calculations during the CMJ. Why have you employed two different operational definitions of the braking phase?

Materials and Methods, line 217: Integration was achieved through trapezoid rule? Please be clear.

Materials and Methods, line 216: Change “works” to “work”. Also lines 218, 220, 221.

Materials and Methods, line 234: Was the ICC model 2-way random or 2-way mixed (trials as fixed effect, participants as random effect)?

Results, Table 1: Close the parentheses around Hedge’s g 95% CI, change Descript to Description (this might have to appear as a note under the table to save space), change Ankl to Ankle.

Results, line 286: Change “works” to “work”. Also line 288, 290, 291, 297, 298, 302, 303, 304, 307, 309.

Discussion, line 315: Change “works” to “work”. Also line 316, 325, 326, 338, 343, 346, 350, 352, 364, 372, 373, 380, 381, 393, 398, 405, 410, 415, 418, 427, 430, and 456.

The English can be improved.

Author Response

Reviewer #1:

The authors investigated the kinematic and kinetic differences between countermovement, drop, and countermovement rebound jumps performed by a large sample of physically active men. The authors also reported the inter-session reliability for the mechanical variables measured. Importantly, the authors conducted an analysis of the positive and negative work performed by the moments at the hip, knee, and ankle joints. The authors have presented information that practitioners will find useful and the manuscript is largely well-written, although there are numerous grammatical errors throughout that require revision and the authors need to ensure that their methods and procedures are clear. Therefore, I am recommending that the authors revise the manuscript based upon the following:

Introduction, line 54: Change “...joint works...” to “...joint work...”

Response: Thanks for the comment, we’ve modified this in revised manuscript.

Introduction, line 67: Change “works” to “work”. Also, note here that an extensor moment is presented as positive by convention, but an extensor moment only does positive work if the motion of the joint is also positive (i.e. a net extensor moment combined with extension at the joint). Consider altering the note presented in the parentheses here.

Response: Thanks for the comment, we have changed the sentence in the parentheses.

Introduction, line 72: Change “A couple of reasons...” to “Variables including the involvement of an arm swing, the countermovement depth...”

Response: Thank you for noting this. we’ve changed this sentence based on reviewer’s comment.

Introduction, line 84: I do not think that you need to include the alpha used in the Holcomb et al. study here as it adds very little. An effect size would be more useful.

Response: Thank you for your comment. We’ve deleted alpha value in the revised manuscript, however, Holcomb et al. in this study didn’t acknowledge the effect size.

Introduction, lines 85-88: You cite the work of Decker and McCaw here, noting that the authors assessed joint work during “40, 60, and 80 cm drop heights.” However, these authors actually had participants drop from a constant height of 60 cm; it was actually the target heights that were manipulated to 40, 60, and 80 cm. Please correct this.

Response: Thanks for pointing out this error, we’ve modified accordingly.

Introduction, line 88: Change “works” to “work”.

Response: We’ve done it, thank you!

Introduction, line 107: Change “works” to “work” (twice).

Response: We’ve done it, thank you!

Materials and Methods, line 128: Change “works” to “work” (twice).

Response: We’ve done it, thank you!

Materials and Methods, line 130: Present a reference for “strategy-based metrics”. Perhaps Bishop et al. (2021).

Response: Thanks for the comment, and a reference from Bishop et al. 2021 has been added.

Materials and Methods, line 159: Present the model number of the force plates used.

Response: Thanks for the comment, and the model number has been added into the parentheses.

Materials and Methods, line 172: You note that a single force plate was used for the DJ. If this is the case, then the calculation of bilateral joint kinetics would not be possible as each leg requires the specific GRF on that side of the body to be measured. Please correct this.

Response: Thanks for the comment. Sorry for the misunderstanding. We did use a twin force plate; here we would like to acknowledge that we used one-force plate method during DJ measures, i.e., landing onto the FP only with no FP been placed on top of the drop box.

Materials and Methods, line 173: Please present the rationale for using a 30 cm box.

Response: Thanks for the comment. We have acknowledged that 30 cm was chosen in accordance with previous DJ investigations.

Materials and Methods, line 193: Presumably joint angular velocities were calculated using first central difference methods? Please add this.

Response: Thank you for your comment. As numerous studies have verified the accuracy and reliability of Visual 3D software regarding the calculation of kinetic and kinematic variables, we only export the joint power data from Visual 3D to calculate the joint work data in MATLAB rather than calculating the joint angular velocities ourselves.

Materials and Methods, line 198: Remove the parenthesis after COM here.

Response: Thank you for noting this, we’ve amended.

Materials and Methods, line 201: You note “the net GRF being lower than five times the net GRF extracted from only the middle part of the flight phase.” Do you mean 5 x the standard deviation of the GRF during flight, or actually 5 x the GRF during flight? See also lines 203-204 and line 205.

Response: Thank you for noting this, we’ve amended into ‘five times the standard deviation of vertical GRF’.

Materials and Methods, lines 206-208: The braking phase here includes the “yielding” phase that is absent from your calculations during the CMJ. Why have you employed two different operational definitions of the braking phase?

Response: Thank you very much for raising this valuable comment. We check our MATLAB code; we do use the same definition in our code but with wrong definition in this manuscript. We now amended the definition of braking phase of DJ.

Materials and Methods, line 217: Integration was achieved through trapezoid rule? Please be clear.
Response: Thank you for this comment, the parentheses contain ‘through trapezoid rule’ has been added.

Materials and Methods, line 216: Change “works” to “work”. Also lines 218, 220, 221.

Response: all done, thank you!

Materials and Methods, line 234: Was the ICC model 2-way random or 2-way mixed (trials as fixed effect, participants as random effect)?

Response: Thanks for your comment. We’ve amended the related sentence.

Results, Table 1: Close the parentheses around Hedge’s g 95% CI, change Descript to Description (this might have to appear as a note under the table to save space), change Ankl to Ankle.

Response: Thanks for taking a note on this. We assume this might be an editorial office problem. Because in our manuscript, it shows ‘Ankle’, ‘Descriptor’ and all ICC and CV have their respective 95% CI values as well.

Results, line 286: Change “works” to “work”. Also line 288, 290, 291, 297, 298, 302, 303, 304, 307, 309.

Response: all done, thank you!

Discussion, line 315: Change “works” to “work”. Also line 316, 325, 326, 338, 343, 346, 350, 352, 364, 372, 373, 380, 381, 393, 398, 405, 410, 415, 418, 427, 430, and 456.

Response: all done, thank you!

Thank you again for your generosity with your precious time invested in improving our submission.

Sincerely,

The Authors

Reviewer 2 Report

The study collects data from a substantial number of subjects and measures kinetic and kinematic factors across two separate sessions to evaluate between-session reliability. Joint mechanical work is calculated as a proportion of total work for each joint during each type of jump. The ankle and knee joints are identified as crucial contributors to total work, indicating a need for focused improvement in the muscles associated with these joints. Moreover, the study underscores the significance of eccentric training, as the knee joint demonstrates higher negative work compared to other joints in CMRJ.

   The article's strengths encompass a well-structured introduction that contextualizes the subject and incorporates pertinent citations. The methods and materials section offers comprehensive details about participants, procedures, and measurements, enhancing the transparency of the experimental setup. The discussion is well-organized, commencing with the study's objectives, presenting the results, and delivering insightful analysis, including discrepancies with prior research findings and plausible explanations for these disparities.

Author Response

Reviewer #2:

The study collects data from a substantial number of subjects and measures kinetic and kinematic factors across two separate sessions to evaluate between-session reliability. Joint mechanical work is calculated as a proportion of total work for each joint during each type of jump. The ankle and knee joints are identified as crucial contributors to total work, indicating a need for focused improvement in the muscles associated with these joints. Moreover, the study underscores the significance of eccentric training, as the knee joint demonstrates higher negative work compared to other joints in CMRJ.

The article's strengths encompass a well-structured introduction that contextualizes the subject and incorporates pertinent citations. The methods and materials section offers comprehensive details about participants, procedures, and measurements, enhancing the transparency of the experimental setup. The discussion is well-organized, commencing with the study's objectives, presenting the results, and delivering insightful analysis, including discrepancies with prior research findings and plausible explanations for these disparities.

Thank you again for your generosity with precious time invested in our paper. We found our submission improved after careful amendment based on your valuable comments.

Sincerely,

The Authors

Reviewer 3 Report

This topics does not deserve a real scientific and practical interest for the readers; furthermore, the methodology as the verbal consign depending on the jump types* does not garanti a real reproducibility of the experiment.

*the specific verbal cues 152 such as “jump as high and as fast as you can” was given for the CMJ and CMRJ1, and 153 “jump as high as you can whilst spending the shortest time possible on the ground” was 154 given for the DJ and CMRJ2. These external verbal cues direct participants’ attention to 155 the movement outcomes only and ensure they utilize self-preferred jump strategies.

Author Response

Reviewer #3:

This topics does not deserve a real scientific and practical interest for the readers; furthermore, the methodology as the verbal consign depending on the jump types* does not garanti a real reproducibility of the experiment.

*the specific verbal cues 152 such as “jump as high and as fast as you can” was given for the CMJ and CMRJ1, and 153 “jump as high as you can whilst spending the shortest time possible on the ground” was 154 given for the DJ and CMRJ2. These external verbal cues direct participants’ attention to 155 the movement outcomes only and ensure they utilize self-preferred jump strategies.

Response: Thank you again for your generosity with precious time invested in identifying our errors regarding clarity and writing. We found our submission improved after careful amendment based on your valuable comments.

Sincerely,

The Authors